# Examining Transcriptomic Alterations in Rat Models of Intracerebral Hemorrhage and Severe Intracerebral Hemorrhage

**DOI:** 10.3390/biom14060678

**Published:** 2024-06-11

**Authors:** Shaik Ismail Mohammed Thangameeran, Sheng-Tzung Tsai, Hock-Kean Liew, Cheng-Yoong Pang

**Affiliations:** 1Institute of Medical Sciences, Tzu Chi University, Hualien 97004, Taiwan; 106324122@gms.tcu.edu.tw (S.I.M.T.); flydream.tsai@gmail.com (S.-T.T.); 2Neuro-Medical Scientific Center, Hualien Tzu Chi Hospital, Buddhist Tzu Chi Medical Foundation, Hualien 97004, Taiwan; 3Department of Neurosurgery, Hualien Tzu Chi Hospital, Buddhist Tzu Chi Medical Foundation, Hualien 97004, Taiwan; 4PhD Program in Pharmacology and Toxicology, Tzu Chi University, Hualien 97004, Taiwan; 5Department of Medical Research, Hualien Tzu Chi Hospital, Buddhist Tzu Chi Medical Foundation, Hualien 97004, Taiwan

**Keywords:** transcriptomics, intracerebral hemorrhage, severe intracerebral hemorrhage, gene expression, neuroinflammation

## Abstract

Intracerebral hemorrhage (ICH) is a life-threatening condition associated with significant morbidity and mortality. This study investigates transcriptomic alterations in rodent models of ICH and severe ICH to shed light on the genetic pathways involved in hemorrhagic brain injury. We performed principal component analysis, revealing distinct principal component segments of normal rats compared to ICH and severe ICH rats. We employed heatmaps and volcano plots to identify differentially expressed genes and utilized bar plots and KEGG pathway analysis to elucidate the molecular pathways involved. We identified a multitude of differentially expressed genes in both the ICH and severe ICH models. Our results revealed 5679 common genes among the normal, ICH, and severe ICH groups in the upregulated genes group, and 1196 common genes in the downregulated genes, respectively. A volcano plot comparing these groups further highlighted common genes, including PDPN, TIMP1, SERPINE1, TUBB6, and CD44. These findings underscore the complex interplay of genes involved in inflammation, oxidative stress, and neuronal damage. Furthermore, pathway enrichment analysis uncovered key signaling pathways, including the TNF signaling pathway, protein processing in the endoplasmic reticulum, MAPK signaling pathway, and Fc gamma R-mediated phagocytosis, implicated in the pathogenesis of ICH.

## 1. Introduction

Intracerebral hemorrhage (ICH) stands as one of the most devastating cerebrovascular events, exacting a profound toll on both individuals and healthcare systems worldwide [1,2]. This hemorrhagic stroke subtype, characterized by the rupture of blood vessels within the brain parenchyma, leads to the abrupt release of blood into the cerebral tissue, culminating in neurological deficits and, frequently, mortality [3,4,5]. Despite advances in medical care, ICH remains a formidable clinical challenge, with limited therapeutic options. Understanding the intricate molecular alterations underpinning ICH and its severe variants is pivotal for the development of effective interventions [6]. ICH accounts for approximately 10–15% of all stroke cases [7]. The condition manifests acutely, with the rapid accumulation of blood in the brain leading to elevated intracranial pressure and secondary damage, notably due to inflammation, oxidative stress, and neuronal toxicity [8,9,10]. Consequently, an in-depth understanding of the molecular underpinnings of ICH is critical for devising therapeutic strategies aimed at mitigating the deleterious consequences of this condition.

In recent years, the advent of high-throughput molecular biology techniques has revolutionized our capacity to scrutinize the transcriptome, allowing for the systematic analysis of gene expression patterns in disease states [11,12]. In the context of ICH, transcriptomic studies provide an opportunity to decipher the intricate gene expression alterations occurring in the hemorrhagic brain and the potential therapeutic targets that emerge from this analysis. RNA sequencing (RNA-seq) has revolutionized molecular biology research by offering precise insights into the dynamic transcriptome. This high-throughput technology surpasses microarrays in gene detection and facilitates the identification of differentially expressed genes (DEGs) using tools like DESeq2 [13,14,15,16]. Furthermore, RNA-seq data analysis is complemented by Kyoto Encyclopedia of Genes and Genomes (KEGG) analysis, Gene Ontology (GO) analysis, Ingenuity Pathway Analysis (IPA), and protein–protein interaction (PPI) analysis, collectively enhancing our understanding of gene function and its implications in various disciplines of molecular biology research [17,18,19].

This study delves into the transcriptomic changes occurring in rodent models of ICH and severe ICH, offering fresh insights into the genetic pathways that govern hemorrhagic brain injury. Our research builds upon previous studies that have explored the transcriptomic landscape of ICH, shedding light on the molecular players involved in its pathophysiology. These investigations have identified genes and pathways associated with inflammation, oxidative stress, and apoptosis in the context of ICH [17,20,21]. While these studies have contributed significantly to our understanding of ICH pathophysiology, several crucial gaps remain. Particularly, the molecular distinctions between ICH and its severe form, characterized by larger hematoma volumes and more profound neurological deficits, have not been comprehensively explored. This study aims to bridge this knowledge gap by scrutinizing the transcriptomic alterations occurring in both ICH and severe ICH, thereby unraveling potential molecular determinants of disease severity.

## 2. Materials and Methods

### 2.1. Animals

All animals were provided with unrestricted access to both food and water and were maintained in a controlled environment with a 12-h light–dark cycle at a temperature of 22 °C. A total of 11 male Sprague–Dawley rats, aged 8 weeks old and weighing between 300–350 g, were used in this study. Only healthy rats, without any observable signs of disease or abnormality, were included in this study. Rats exhibiting any pre-existing health conditions or abnormalities during the initial screening were excluded. Additionally, rats that encountered complications during the surgical procedures for inducing ICH and severe ICH, those showing severe postoperative complications such as excessive bleeding, and those that did not survive the initial postoperative period were excluded from the final analysis. All experimental procedures strictly adhered to the guidelines outlined in the National Institute of Health Guidelines for the Care and Use of Laboratory Animals. All animal experiment protocols were approved by the Institutional Animal Care and Use Committee (IUCAC) at Hualien Tzu Chi Hospital (Approval No. IUCAC 109-78-R1), Taiwan. Every possible measure was taken to minimize any potential distress or suffering experienced by the animals during their handling and humane euthanasia.

### 2.2. Intracerebral Hemorrhage Model

Stereotaxic surgery to induce ICH was carried out by injecting bacterial collagenase VII-S (0.23 U in 1.11 μL sterile saline) into the right striatum of the rat (0.0 mm posterior, 3.0 mm right, 5.0 mm ventral to bregma from the surface of the skull) [22,23,24]. The induction of severe ICH in rats was achieved through a stereotaxic surgical procedure, which involved the injection of bacterial collagenase VII-S (0.6 U in 3 μL sterile saline) into the right striatum (0.0 mm posterior, 3.0 mm to the right, and 5.0 mm ventral to the bregma, as measured from the skull’s surface) [25].

### 2.3. Morphometric Measurements

Hematoma volume was quantified using morphometric measurements. After the induction of ICH in rats, their brains were carefully removed and sectioned coronally through the needle entry site. Serial brain slices of 2 mm thickness were obtained both anteriorly and posteriorly to the needle entry site. These slices were then imaged using a digital camera. The images of the serial slices with visible blood clots were analyzed using ImageJ software (1.43 release, NIH, Bethesda, MD, USA). The total volume of the hematoma (mm^3^) was calculated by multiplying the area of the hematoma in each slice by the distance between the sections that contained the hematoma. This method allowed for an accurate and reproducible assessment of the hematoma volume in each animal [22,26].

### 2.4. RNA Sequencing and Data Analysis

In this study, we utilized “https://usegalaxy.eu/ (accessed on 5 April 2022)”, a comprehensive and user-friendly platform, for the RNA sequencing data analysis of rat models in ICH and severe ICH.

#### 2.4.1. Library Preparation and Sequencing

Each brain tissue was assigned a unique index code. Subsequently, RNA was reverse transcribed into cDNA, and cDNA fragments within the 250–300 bp range were selectively enriched using the kit used. Amplification by polymerase chain reaction (PCR) was performed on the cDNA, followed by purification of the resulting PCR products using the AMPure XP magnetic beads (Beckman Coulter, Beverly, CA, USA). The quality of the sequencing library was assessed using the Agilent BioAnalyzer 2100 system (Santa Clara, CA, USA) for fragment size and the Real-Time PCR system for library concentration, and the library was prepared for paired-end sequencing with read lengths of 125/150 bp.

#### 2.4.2. Sequencing

The prepared libraries were sequenced on the Illumina NovaSeq 6000 system (San Diego, CA, USA) with a 150 PE sequencing format to generate 150 bp paired-end reads. Sequencing was performed to achieve a minimum depth of 30 million reads per sample, ensuring comprehensive coverage and depth for accurate differential expression analysis.

### 2.5. RNA Sequencing Data Analysis

The RNA sequencing data were analyzed via https://usegalaxy.eu/ [27,28,29], adhering to a sequence of quality assurance and data processing steps.

#### 2.5.1. Quality Control with FastQC and MultiQC

Raw sequencing reads underwent initial quality assessment using FastQC (v0.11.9) [30,31], revealing high-quality score metrics with a mean quality score consistently above 30 across all bases, indicative of excellent sequencing quality. The GC content showed a normal distribution centered around 50%, which is typical for a diverse genomic sample. Sequence duplication levels were acceptable, characterized by a substantial proportion of unique sequence counts compared to duplicates, as visualized in the sequence counts plot. MultiQC (v1.9) [32] compiled these results into a cohesive report, confirming the high-quality status across all sequencing batches.

#### 2.5.2. Read Trimming with Cutadapt and Quality Reassessment

Cutadapt (v2.10) [33] was employed to trim adapters and low-quality bases, configured to remove sequences below a quality threshold of 20 and shorter than 20 base pairs. Following the trimming process, 98% of read pairs were retained, signifying the efficiency of the procedure. The post-trimming analysis revealed a slight reduction in the total base pairs processed, with 97% of the original base pairs being written to the output files. The quality of reads post-trimming was confirmed to be enhanced, as demonstrated by the high retention rate of the sequences and the significant number of base pairs preserved.

#### 2.5.3. RNA Annotation and Alignment with RNA STAR

The RNA STAR aligner (v2.7.8a) [34] was employed for aligning sequencing reads against the rat reference genome (Genome assembly Rnor_6.0), significantly aided by pre-alignment RNA annotation for precise genome mapping. Across various samples, the average mapping rate observed was 62.11%, with the percentage of reads uniquely aligned to the genome averaging 58.74% to 64.94%, highlighting the data’s high specificity. The efficiency in splicing junction discovery varied across samples, reflecting the aligner’s capability to identify both known and novel junctions, which are essential for understanding the transcriptomic complexity.

#### 2.5.4. Gene Expression Quantification and Read Orientation Analysis

Infer Experiment and featureCounts analyses provided critical insights into read orientation and gene expression quantification, respectively. The Infer Experiment [35] analysis confirmed the correct orientation of reads according to the library’s strand-specificity, which is essential for accurate downstream quantification. Subsequently, featureCounts (v2.0.1) [36] quantified gene expression, revealing an average of 1,140,056 reads per gene, with an assignment rate of 33.2% to 36.9% of reads to genomic features. This robust quantification underscores the comprehensive coverage achieved, with a significant portion of genes detected across all samples, illustrating the depth and breadth of our transcriptomic analysis.

#### 2.5.5. Differential Expression Analysis with DESeq2

Using DESeq2 (v1.30.0) [16], the analysis identified 5528 DEGs between conditions at an adjusted *p*-value < 0.05, with 2723 genes upregulated and 2805 genes downregulated. The analysis revealed a fold change range from −3.012 to 7.539 among significant DEGs.

#### 2.5.6. Annotation of DESeq2 Results

The annotated DESeq2 analysis elucidated the biological significance of DEGs, highlighting a subset of 25,358 genes.

### 2.6. Functional and Pathway Enrichment Analysis

To explore the roles and relationships of DEGs, we applied GO for functional insights and the KEGG for pathway mapping. We analyzed DEGs using the DAVID tool (version 6.8, medium stringency, https://david.ncifcrf.gov/ accessed on 5 April 2022) to identify significant GO categories and KEGG pathways (*p* < 0.05) [37].

### 2.7. Visualization of Gene Expression

Using SR Plot, we generated a precise heatmap to visualize gene expression patterns, revealing distinct clusters and trends critical for understanding biological processes [38].

### 2.8. Protein-Protein Interaction Analysis (STRING Database)

Protein-protein interaction (PPI) networks were predicted using the Search Tool for the Retrieval of Interacting Genes/Proteins (STRING) database (https://string-db.org/ accessed on 5 April 2022) [39]. STRING integrates known and predicted associations, encompassing direct (physical) and indirect (functional) connections. The input data comprised a list of protein-coding gene symbols, and interaction searches were conducted using a high confidence score threshold of 0.7 to ensure the relevance and reliability of the predicted interactions. The networks were visualized using STRING’s graphical representation, where nodes represent proteins and edges represent the interactions.

## 3. Results

### 3.1. Transcriptome Profiling Reveals Consistent Expression Pattern

The hematomas caused by the collagenase infusion model between rat models of ICH and severe ICH are shown in Figure 1A, and the differences in the hematoma volume are quantitated (Figure 1B). The difference in hematoma volume between ICH and severe ICH is characterized by a significantly larger hematoma volume, approximately threefold greater than that of the ICH model, as illustrated in Figure 1B (Appendix A).

In this study, we employed transcript per million (TPM) as a robust method to quantify gene expression levels across our experimental samples. Using the FeatureCounts tool, we quantified gene expression levels in our dataset. TPM values were obtained, providing a standardized measure of gene expression that facilitated meaningful comparisons across different experimental conditions and datasets [40]. To evaluate the transcriptional response across different samples, we performed RNA sequencing and calculated the transcripts per million distribution to allow for direct comparison across samples [41,42]. As illustrated in Figure 1C, the median log2 TPM values were consistent across the 12 samples [normal (N), I, and I3], indicating no significant global shifts in transcript abundance attributable to our experimental conditions.

The variability within each sample, assessed as the range of log2 TPM values, remained consistent across the board. This uniformity suggests that the conditions of our experiment or the biological differences between our samples did not introduce broad-scale changes in gene expression [43].

### 3.2. Principal Component Analysis of Gene Expression Profiles

Following the comprehensive assessment of transcriptomic responses measured in TPM (Figure 1C), we further investigated the multidimensional data structure using Principal Component Analysis (PCA, Figure 1D). As depicted in Figure 2, PCA of TPM data revealed distinct clustering of the samples corresponding to the experimental conditions. The first principal component (PC1), which accounted for 61.3% of the total variance, effectively separated the condition of normal rat brains (N) from the ICH (I) and the severe ICH (I3), signifying a pronounced divergence in the gene expression profile. Conversely, the overlap between “I” and “I3” along PC2, encompassing 22.8% of the variance, implies potential shared biological pathways, with distinct nuances unique to each condition. These findings are consistent with previous studies that utilized PCA to delineate distinct genomic signatures across different biological conditions [44,45]. The distinct separation among the conditions underscores the robustness of the observed transcriptomic changes and supports the hypothesized biological impact of these conditions on gene expression patterns.

### 3.3. Comparative Transcriptomic Analysis Reveals Differential Gene Expression in ICH and Severe ICH

To elucidate the genetic alterations associated with ICH and its progression to a severe state, we conducted a comprehensive transcriptomic analysis. This analysis involved comparing gene expression profiles of brain tissues from normal, ICH, and severe ICH animals. The results were plotted in Venn diagrams (Figure 1D) to visualize the overlap and unique gene expression changes across these states.

By analyzing and reporting on the differential expression of a large number of genes (9028 upregulated in ICH vs. normal, 12,037 in severe ICH vs. ICH, and 9742 in severe ICH vs. normal), we aim to provide a broad overview of the complex transcriptomic changes that occur in response to ICH and its progression to severe ICH. This comprehensive approach allows us to capture the full scope of biological processes affected by ICH, including those that may not be immediately obvious but could be critical for understanding the disease mechanism and identifying novel therapeutic targets. Additionally, our analysis identified a significant core of 5679 genes commonly upregulated in both ICH and severe ICH compared to normal brains, suggesting a shared response mechanism regardless of ICH severity. Furthermore, we found 1578 genes uniquely upregulated in the transition from ICH to severe ICH, indicating specific genetic responses associated with disease exacerbation.

Conversely, the analysis of downregulated genes revealed 7875 genes in the ICH compared to the normal brains, and 3698 genes in the severe ICH compared to the ICH brains, with 7158 genes downregulated in the severe ICH compared to the normal brains. Of these, 4188 genes were consistently downregulated across all comparisons, pointing towards a sustained suppression of these genes in the context of ICH and its severe form. Furthermore, the transition from the ICH to the severe ICH was marked by the unique downregulation of 911 genes, highlighting a distinct downregulatory gene expression pattern associated with severe ICH.

### 3.4. Differential Gene Expression in Intracerebral Hemorrhage: A Hierarchical Clustering Approach

In this initial heatmap exploration (Figure 2), transcriptional profiles from three distinct cohorts are analyzed: the normal group (N-1, N-2, N-3), the ICH group (I-1, I-2, I-3, I-4), and the severe ICH group (I3-1, I3-2, I3-3, I3-4). This heatmap employs a hierarchical clustering algorithm to delineate the patterns of differential gene expression, thereby elucidating the transcriptional disparities between the normal and pathological states. Extending the analysis, a detailed examination of the normal group establishes a foundational expression profile, which serves as a comparative baseline for the ICH and the severe ICH cohorts. The heatmap distinctly illustrates a notable divergence in gene expression levels in both the ICH and the severe ICH groups relative to the normal group, highlighting a perturbation or modification in the gene regulatory frameworks consequent to hemorrhagic incidents [46].

The ICH group demonstrates a gene expression profile that significantly differs from the normal group, with this contrast becoming more pronounced in the severe ICH group. Such findings suggest a spectrum of transcriptional changes that correspond with the gravity of the hemorrhage. The progressive intensification of gene expression variations from ICH to severe ICH highlights the potential for a proportional relationship between the magnitude of hemorrhagic influence and cellular transcriptional responses.

This preliminary summary establishes the groundwork for further detailed studies into the specific genes and pathways involved in ICH and its advancement to a severe condition. Future research, aimed at providing gene annotations and functional profiles, will shed light on the underlying molecular processes and could reveal potential biomarkers useful for predicting and treating ICH [47].

### 3.5. Differential Gene Expression Analysis via Volcano Plotting

To elucidate the transcriptional changes associated with ICH, we conducted a comparative gene expression analysis among the normal, ICH, and severe ICH brain tissues. The differential expression analysis aimed to identify key genes that were upregulated or downregulated in response to the severity of ICH [30]. Volcano plots were employed as a graphical method to display the results of these comparisons, providing a clear visualization of the relationship between fold change and the statistical significance of gene expression differences (Figure 3) [48,49].

The first volcano plot (Figure 3A) compares normal with the ICH groups, revealing distinct gene expression profiles between the two conditions. A notable number of genes were found to be significantly upregulated in ICH rats [47], as depicted by data points skewed to the right of the plot. Conversely, a subset of genes displayed a downregulation trend, positioned to the left [48]. Genes that surpassed the threshold for statistical significance (adjusted *p*-value) ascended on the y-axis, with those reaching the topmost region being the most significantly altered [49].

Moving to the comparison between ICH and severe ICH (I3) rats, the second volcano plot (Figure 3B) indicates a further shift in gene expression with disease progression. Several genes that were not differentially expressed in the ICH versus normal comparison emerged as significantly modulated when the severity of ICH increased. This change implies a potential dose–response relationship between gene expression and ICH severity [50,51].

The third plot (Figure 3C), representing the comparison between normal rats and those with severe ICH, provided the most dramatic illustration of gene expression alterations. The substantial number of genes upregulated or downregulated in the severe ICH context underscores the profound transcriptional impact of ICH severity on the rat genome [52].

Key genes identified from the plots were annotated with their respective fold change and *p*-value, highlighting those with the most pronounced expression differences. For instance, genes such as Timpg1 and Serpine1 appear prominently in the ICH versus normal comparison, suggesting their potential roles in the pathophysiological response to ICH [53,54].

Complementing our volcano plot findings, Venn diagrams were utilized to compare upregulated and downregulated genes across N, I, and I3 rat groups. A significant overlap of 5679 upregulated and 4188 downregulated genes across all conditions suggests a core set of genes consistently associated with ICH. Unique gene expression signatures were also identified in each comparison, reflecting the nuanced genetic shifts that correspond to the progression and severity of ICH. The results from these volcano plots and Venn diagrams pave the way for a more detailed Gene Ontology analysis, which will further dissect the biological significance of these differentially expressed genes. Through this subsequent analysis, we anticipate uncovering the biological processes, cellular components, and molecular functions most perturbed by ICH [55].

### 3.6. KEGG and GO Enrichment in Normal vs. ICH

#### 3.6.1. Upregulated Gene Pathways in ICH

Our Kyoto Encyclopedia of Genes and Genomes (KEGG) and Gene Ontology (GO) analyses indicate a significant upregulation of genes in pathways associated with basal metabolic processes in ICH, specifically in basal transcription factors and amino acid degradation. The increased activity within basal transcription factors may represent a cellular attempt to enhance gene expression and restore homeostasis following hemorrhagic stress [56]. The upregulation of amino acid degradation pathways, particularly those involving branched-chain amino acids like valine, leucine, and isoleucine, underscores a shift toward catabolism, possibly to meet heightened energy demands during ICH [57,58,59]. As severity escalates to severe ICH, our data reveal an additional upregulation of genes involved in the spliceosome and homologous recombination—pathways crucial for DNA repair and RNA processing. This suggests an enhanced repair mechanism activation, likely a response to increased DNA damage and the need for accurate RNA splicing under severe stress conditions (Figure 4) [60,61].

#### 3.6.2. Downregulated Gene Pathways in ICH

In contrast, pathways associated with inflammatory responses and apoptosis are downregulated in ICH. This includes the TNF signaling pathway, which is typically involved in mediating inflammatory responses and can exacerbate tissue damage when overactivated. Its downregulation may thus represent a protective, anti-inflammatory response post-ICH [62,63]. Additionally, the observed downregulation in apoptosis pathways might reflect a cellular survival strategy aimed at limiting cell death in the acute phase of hemorrhagic injury [64,65,66]. The severity of ICH appears to further influence downregulated pathways, with severe ICH showing reduced expression in pathways critical for neuronal communication, like axon guidance and glutamatergic synapse. The suppression of these pathways could contribute to the neurological impairment observed in severe ICH cases, indicating a loss of synaptic integrity and axonal connectivity (Figure 5) [67,68,69].

### 3.7. KEGG and GO Enrichment in ICH vs. Severe ICH-Upregulated Gene Pathways in Severe ICH

#### 3.7.1. Focused Upregulation in Cellular Recovery Pathways

Among the upregulated pathways, the Homologous Recombination (HR) pathway stands out, signifying an activation of the intricate DNA repair machinery. The HR pathway, indicated by the red blocks in the KEGG pathway diagrams, shows a considerable increase in the expression of genes like BRCA1 and RAD51, which are quintessential for the repair of double-strand breaks and the preservation of genomic stability under the duress of hemorrhagic stress. This elevation is indicative of the cell’s propensity to counteract the DNA damage and maintain chromosomal integrity (Figure 6A,B).

#### 3.7.2. Ubiquitin-Mediated Proteolysis Pathway Amplification

In the setting of severe ICH, the pathway of Ubiquitin-mediated Proteolysis exhibits significant gene upregulation, illustrated by the intensified red blocks on the pathway diagram. The enhanced expression of genes such as HSPA5 and EDEM1 implies an intensified focus on protein quality control, particularly the selective degradation of misfolded or damaged proteins. The pathway’s activation indicates the endoplasmic reticulum’s vital role in managing proteotoxic stress, underscoring the importance of the ubiquitin–proteasome system in ensuring protein homeostasis during cellular crises [70] (Figure 6C).

#### 3.7.3. Pathways of Neurodegeneration

In the context of severe ICH, our KEGG pathway analysis demonstrates a remarkable upregulation of neurodegenerative pathways. This is particularly evident in the Alzheimer’s disease pathway, where key genes involved in amyloid processing and tau protein pathology, such as APP and MAPT, exhibit significant increases in expression. These genes, highlighted by the pronounced red indicators, suggest an intensified cellular response to neurodegenerative stimuli in severe ICH, potentially exacerbating the disease progression (Figure 6D).

### 3.8. KEGG and GO Enrichment in ICH vs. Severe ICH-Downregulated Gene Pathways in Severe ICH

#### 3.8.1. Downregulation of Phagocytosis Pathway

Prominent among the downregulated pathways is the Fc gamma R-mediated phagocytosis pathway, as evidenced by the red blocks in the KEGG pathway diagrams. Key components such as FCGRs, which play a pivotal role in immune response and phagocytosis, exhibit reduced expression. This suggests a potential impairment of the innate immune system’s ability to clear cellular debris and pathogens in severe ICH, possibly exacerbating the inflammatory response and contributing to secondary brain injury (Figure 7A,B).

#### 3.8.2. Impaired Cell Growth and Communication

The Ras signaling pathway, another critical cascade for cell growth and communication, is notably suppressed in severe ICH. The red blocks highlight downregulated genes such as KRAS, suggesting a disruption in vital signaling that could affect cell survival and neuroplasticity after hemorrhagic events. This pathway’s suppression may be indicative of a broader impairment in cell signaling mechanisms that are essential for recovery and regeneration in the brain (Figure 7C).

### 3.9. STRING Pathway Analysis

In the present study, we performed a comprehensive STRING pathway analysis to decipher the intricate web of protein–protein interactions (PPIs) within a selection of 50 genes. These genes were chosen based on their differential expression between the normal, ICH, and severe ICH groups.

The differentially expressed genes between the normal and the ICH groups showed a high degree of connectivity and potential functional significance. For instance, genes such as Ccl2, Cxcl2, and Ccl7, which are known chemokines, showed a dense cluster of interactions, suggesting a potent inflammatory response in the ICH group compared to the normal group. Additionally, the matrix metalloproteinase gene, MMP9, and the heat shock protein, Hspb1, were found to be central hubs, indicating their crucial roles in post-ICH responses, possibly mediating processes such as matrix remodeling and cellular stress responses, respectively (Figure 8A).

Interconnecting the genes from the normal and the severe ICH revealed a significant divergence in the molecular profiles of the conditions. Central to the severe ICH network was the upregulation of S100a9, a calcium-binding protein, often associated with acute and chronic inflammation, which serves as a nodal point in the network, suggesting a state of heightened inflammatory response in severe ICH. Another prominent feature was the connectivity around Nos2, indicative of increased nitrosative stress during severe ICH. The interactions involving Nos2 may implicate this enzyme in the broader context of cellular damage and neuroinflammation seen in severe ICH. Additionally, our analysis highlighted F7 and Serpine1 as key players, with their enhanced interactions suggesting a shift towards a pro-thrombotic and anti-fibrinolytic state during severe ICH. These alterations in the hemostatic gene network could contribute to the worsened prognosis often associated with severe ICH (Figure 8B). The chemokines Cxcl2 and Ccl2 also exhibited pronounced connectivity, reinforcing the notion of an aggressive inflammatory response during severe ICH. This is further corroborated by the central positioning of Hspb1 within the network, which may reflect a cellular attempt to counteract the increased stress conditions present in severe ICH through the upregulation of molecular chaperones.

The differentially expressed genes between the ICH and the severe ICH have elucidated several noteworthy alterations in the PPI landscape when advancing from ICH to severe ICH. Notable among these is the emergence of Inhba, a member of the TGF-β superfamily, which showed enhanced interactions in the severe ICH state. Its central role in the network suggests an upregulated involvement in processes such as fibrosis, inflammation, and cellular proliferation in the context of severe ICH (Figure 8C).

An intriguing observation was the increased connectivity of Wnt1 in severe ICH, pointing to its potential role in modulating neuroinflammatory pathways and possibly contributing to the secondary injury cascades that exacerbate hemorrhagic damage. Similarly, the gene Adam33, known to be involved in tissue remodeling, displayed heightened interactions, which could reflect the intensified reparative or destructive processes occurring in severe ICH.

Furthermore, the analysis revealed a complex cluster involving Cxcl13, Gpr111, and Galr1, suggesting a heightened chemotactic response and altered neurotransmission, respectively, in severe ICH. The positioning of these genes indicates their possible contribution to the exacerbated inflammatory response and dysregulated neuronal signaling associated with severe ICH.

## 4. Discussion

Our study embarked on a comprehensive transcriptomic analysis to elucidate the molecular underpinnings of ICH and its severe form in rat models. The analysis identified several genes and pathways that were differentially expressed in response to ICH and severe ICH, providing insights into the complex biological processes involved in these conditions.

Key genes such as PDPN, TIMP1, SERPINE1, TUBB6, and CD44 were found to be significantly altered. PDPN, known for its role in lymphatic vessel formation and function, showed notable upregulation. This finding aligns with the hypothesis that vascular abnormalities play a critical role in ICH pathology. TIMP1 and SERPINE1, both involved in extracellular matrix remodeling and inflammation, were also upregulated, suggesting their involvement in the post-ICH inflammatory response and tissue repair processes [21,71,72,73,74,75,76,77,78,79].

Additionally, our analysis highlighted the activation of specific pathways related to inflammation, immune response, and cell proliferation. This is particularly evident in the severe ICH model, where the expression patterns suggest an amplified inflammatory and immune response, possibly contributing to the increased severity and poorer prognosis observed in these cases [80].

The identified genes and pathways have been previously implicated in ICH, but their specific roles and interactions remain not fully understood. For instance, the upregulation of PDPN has been observed in other studies focusing on brain injuries, hinting at a potential universal response mechanism to cerebral damage [72]. Similarly, the roles of TIMP1 and SERPINE1 in inflammation and tissue remodeling have been documented in various neurological disorders, but their specific contributions to ICH progression are less explored [77,81].

The differential expression of genes like PDPN, TIMP1, SERPINE1, TUBB6, and CD44 in our study provides pivotal insights into the pathophysiological processes of ICH and its severe form. The upregulation of PDPN, for instance, suggests a potential involvement in vascular changes post-ICH [75], possibly contributing to hemorrhagic expansion or edema formation. This aligns with recent studies that have emphasized the role of vascular dysfunction in ICH pathology [80].

Furthermore, the observed upregulation of genes involved in inflammation and tissue remodeling, such as TIMP1 and SERPINE1, indicates an active inflammatory response and repair mechanisms following ICH. This is consistent with the notion that inflammation plays a dual role in ICH, both contributing to secondary injury and facilitating recovery. The balance between these opposing processes could be key in determining the outcome after ICH [9,82,83].

Moreover, the pronounced expression changes in the severe ICH model underscore the escalated inflammatory and immune responses in more severe cases. This might provide an explanation for the observed differences in clinical outcomes between standard ICH and its severe form. Understanding these differences at the molecular level could aid in developing targeted therapies that modulate these pathways, potentially improving outcomes for patients with severe ICH [84].

## 5. Conclusions

In conclusion, our study represents a significant step forward in understanding the transcriptomic landscape of ICH and its severe form. The differential expression of key genes and pathways provides valuable insights into the molecular mechanisms underlying ICH pathology. These findings not only enhance our understanding of the disease but also pave the way for future research aimed at translating these insights into clinical applications. As we continue to unravel the complexities of ICH, it is our hope that this research will contribute to the development of more effective diagnostic and therapeutic strategies for this debilitating condition.

## 6. Strengths and Limitations

One of the strengths of our study is the use of comprehensive transcriptomic analysis, which allowed for a broad overview of the genetic profiles during ICH. This approach enabled us to uncover novel genes and pathways that might play significant roles in the disease process. Additionally, the utilization of advanced bioinformatics tools provided a robust analysis of the complex data sets. The severe ICH animal model is the second strength of this study: it allowed us to investigate the candidates behind the more fatal scenario of ICH.

However, our study is not without limitations. The use of a rat model, while providing valuable insights, might not fully replicate the human condition. Differences in species-specific gene expression and responses to injury could influence the translatability of our findings to human ICH. Furthermore, the sample size in our study was limited, which might affect the generalizability of our results. Future studies with larger sample sizes and, ideally, human subjects are necessary to validate and extend our findings.

## 7. Future Directions

Our findings open several avenues for future research. Firstly, there is a need to further explore the roles of PDPN, TIMP1, SERPINE1, TUBB6, and CD44 in the context of ICH. Studies focusing on the mechanistic aspects of these genes could provide deeper insights into their specific functions and interactions in ICH pathology.

Additionally, translating these findings into clinical practice is a critical next step. Investigating the potential of these gene expressions as biomarkers for ICH severity or as therapeutic targets warrants attention. For instance, modulation of the inflammatory response, guided by the expression profiles of TIMP1 and SERPINE1, could be a promising therapeutic strategy.

Furthermore, expanding the research to include human subjects and larger sample sizes will be essential to validate and refine our understanding. Comparative studies between animal models and human ICH cases would also help in elucidating species-specific differences and similarities in the transcriptomic response to ICH.

Finally, integrating other omics data, such as proteomics and metabolomics, with our transcriptomic findings could offer a more comprehensive view of the molecular landscape of ICH. This multidimensional approach might reveal new interactions and pathways that are crucial in the disease process.

## Figures and Tables

**Figure 1 biomolecules-14-00678-f001:**
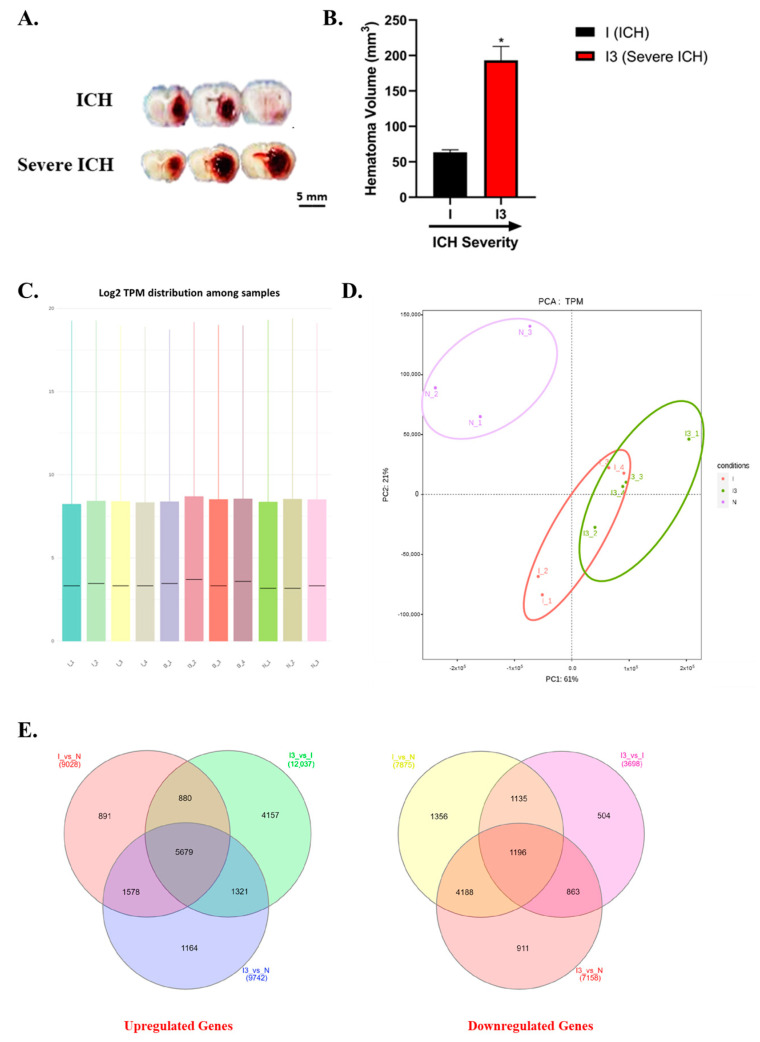
Comprehensive analysis of intracerebral hemorrhage severity and transcriptomic changes. Representative images of brain slices from ICH and severe ICH are shown (**A**). The volume of the striatal hematomas formed in the ICH (I) and severe ICH (I3) is quantitated (**B**); volume in mm^3^ for I and I3 (* *p* < 0.05). The hematoma in the severe ICH rats is nearly 3-fold larger than that in the ICH rats. Box plots illustrating the distribution of log2-transformed transcripts per million (TPM) (**C**). The Principal Component Analysis (PCA) scatter plot depicts the segregation of transcriptomic profiles between the brain tissue from normal (N), ICH (I), and severe ICH (I3) animals (**D**). The ellipses (purple: N, red: ICH, and green: severe ICH) represent a 95% confidence interval for the dispersion of the conditions in the multidimensional space. Venn diagrams (**E**) showing the overlap of differentially expressed genes between conditions: (**left**) upregulated genes in I versus N, I3 versus I, and I3 versus N, respectively; (**right**) downregulated genes in the same comparative groups (*n* = 3 for N and *n* = 4 for I and I3, respectively).

**Figure 2 biomolecules-14-00678-f002:**
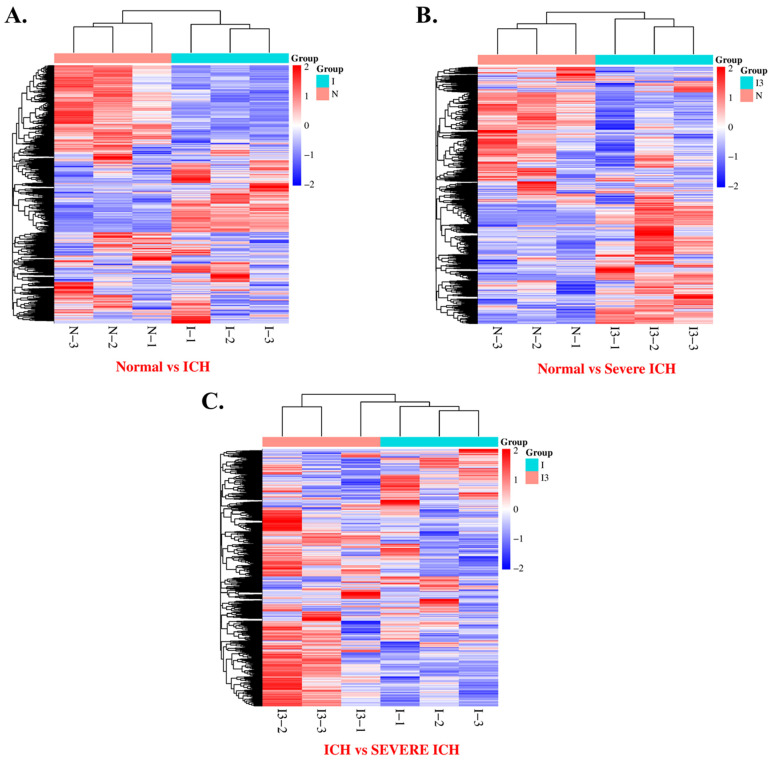
Heatmaps representing the differential expression profiles in pairwise comparisons of transcriptomic data from normal (N) vs. ICH (I) (**A**), normal (N) vs. severe ICH (I3) (**B**), and ICH (I) vs. severe ICH (I3) (**C**), respectively, are shown. The criterion for identification of DEGs is FC > 1.5 or FC < 0.67, *p* < 0.05.

**Figure 3 biomolecules-14-00678-f003:**
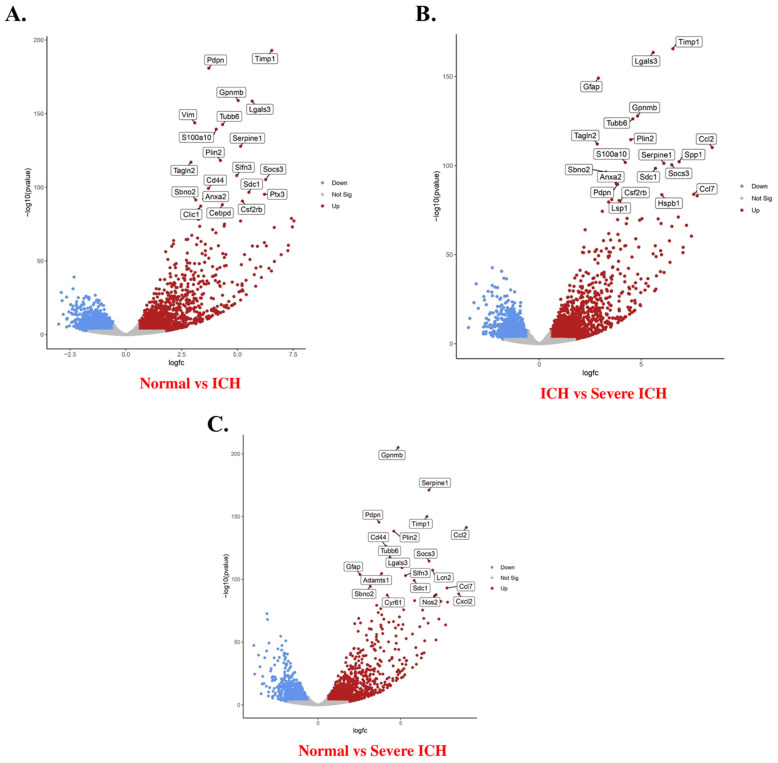
Volcano plots of differential gene expression analysis visualizing different pairwise expressions of gene expression data. The criterion for identification of DEGs is FC > 1.5 or FC < 0.67, *p* < 0.05. (**A**) normal vs. ICH, (**B**) ICH vs. severe ICH, (**C**) normal vs. severe ICH.

**Figure 4 biomolecules-14-00678-f004:**
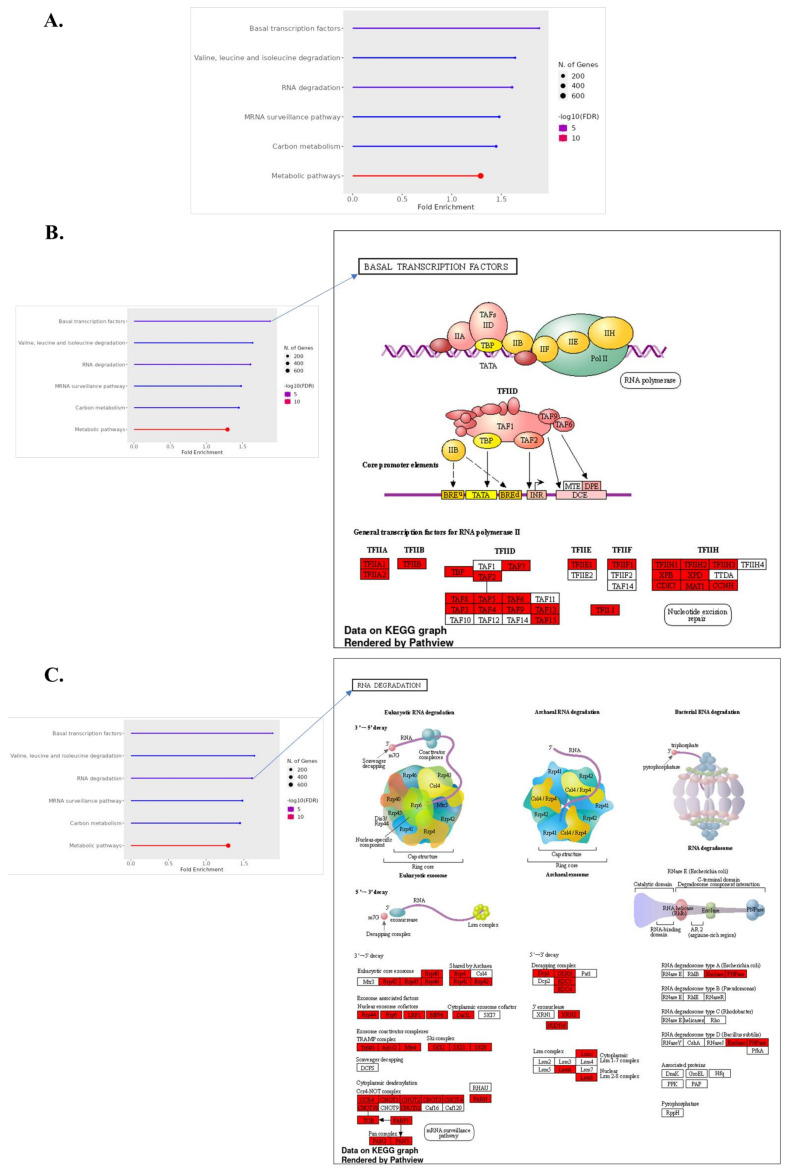
Functional enrichment analysis of upregulated and downregulated genes in normal vs. ICH vs. severe ICH conditions, identified through KEGG pathway enrichment analysis. Bar plot indicating the fold enrichment of selected pathways, with the number of genes involved depicted by the size of the dots and the false discovery rate (FDR) represented by color intensity (**A**). Detailed KEGG pathway map for “Basal Transcription Factors” (**B**) and “RNA Degradation” (**C**). Upregulated genes during ICH (normal vs. ICH) with an FC > 1.5 are colored in red in (**B**,**C**).

**Figure 5 biomolecules-14-00678-f005:**
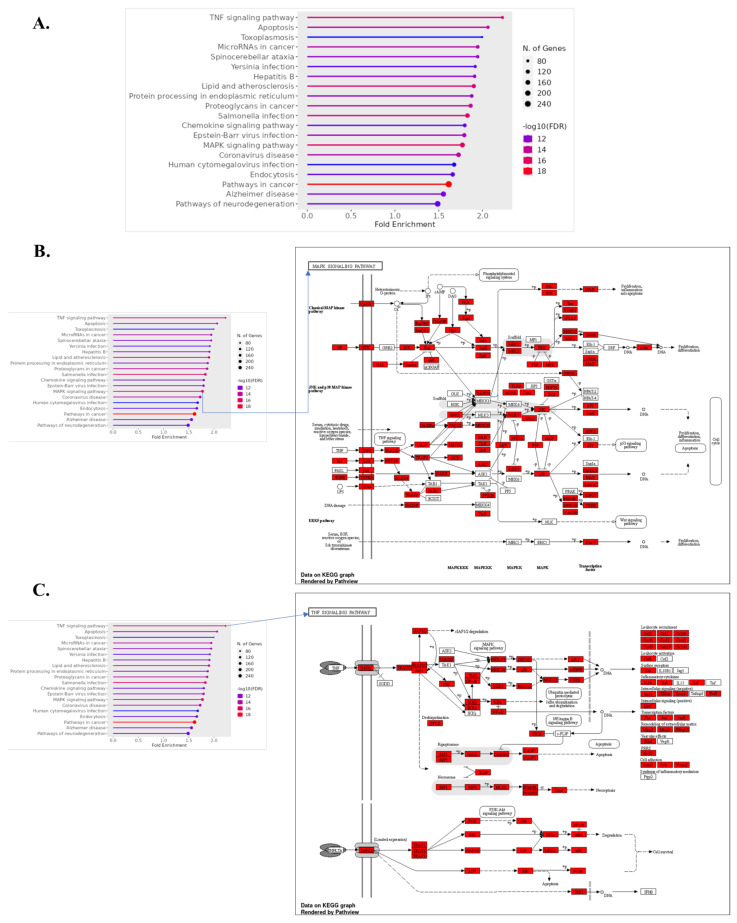
Functional enrichment analysis of upregulated and downregulated genes in normal vs. ICH vs. severe ICH conditions identified through KEGG pathway enrichment analysis. Bar plot indicating the fold enrichment of selected pathways, with the number of genes involved depicted by the size of the dots and the false discovery rate (FDR) represented by color intensity (**A**). Detailed KEGG pathway map for “MAPK Signaling Pathway” (**B**) and “TNF Signaling Pathway” (**C**). Downregulated genes during ICH (normal vs. ICH) with an FC < 0.67 are colored in red.

**Figure 6 biomolecules-14-00678-f006:**
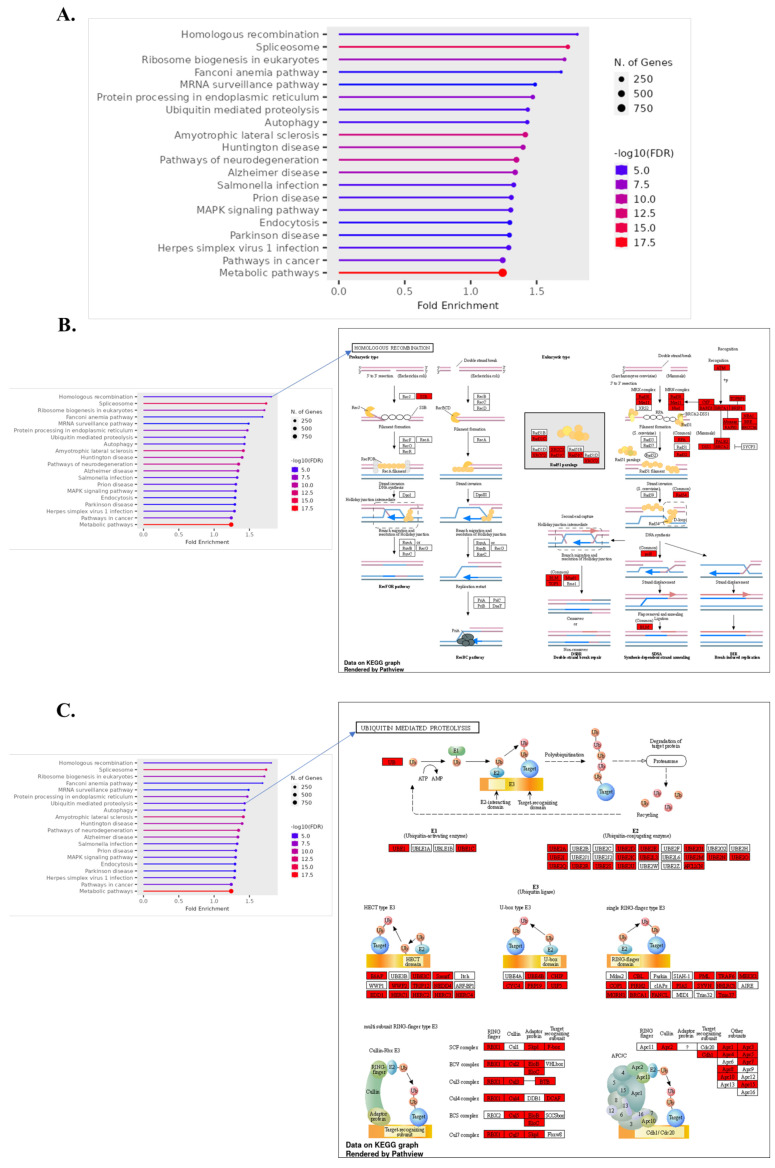
Functional enrichment analysis of upregulated and downregulated genes in normal vs. ICH vs. severe ICH conditions, identified through KEGG pathway enrichment analysis. Bar plot indicating the fold enrichment of selected pathways, with the number of genes involved depicted by the size of the dots and the false discovery rate (FDR) represented by color intensity (**A**). Detailed KEGG pathway map for “Homologous Recombination” (**B**), “Ubiquitin Mediated Proteolysis” (**C**), and “Pathways of Neurodegeneration-Multiple Diseases” (**D**). Upregulated genes during severe ICH (ICH vs. severe ICH) with an FC > 1.5 are colored in red.

**Figure 7 biomolecules-14-00678-f007:**
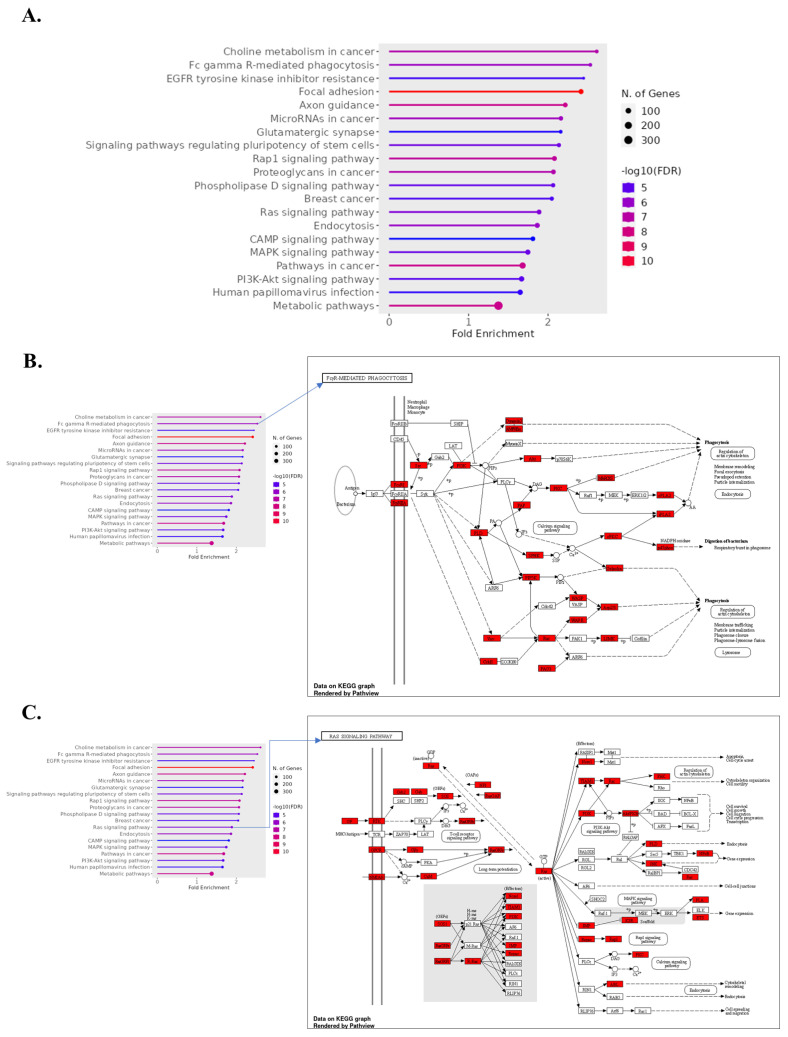
Functional enrichment analysis of upregulated and downregulated genes in normal vs. ICH vs. severe ICH conditions, identified through KEGG pathway enrichment analysis. Bar plot indicating the fold enrichment of selected pathways, with the number of genes involved depicted by the size of the dots and the false discovery rate (FDR) represented by color intensity (**A**). Detailed KEGG pathway map for “FcγR-Mediated Phagocytosis” (**B**) and “RAS Signaling Pathway” (**C**). Downregulated genes during severe ICH (ICH vs. severe ICH) with an FC < 0.67 are colored in red.

**Figure 8 biomolecules-14-00678-f008:**
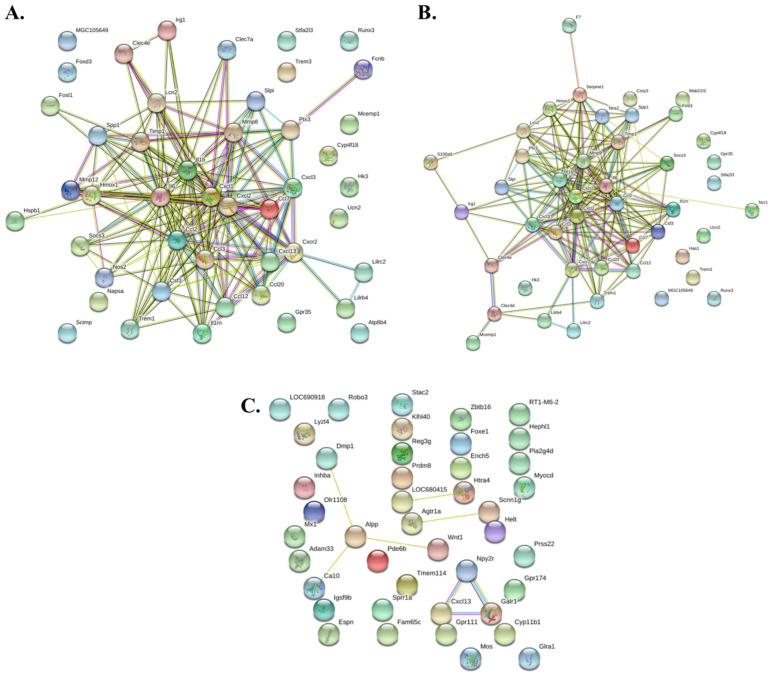
STRING pathway analysis depicting protein–protein interaction (PPI) networks. (**A**) PPI network for ICH vs. normal; (**B**) PPI network of severe ICH vs. normal; and (**C**) PPI network of severe ICH vs. ICH. The analysis parameters were set with a high-confidence score threshold (0.7) to ensure the specificity of interactions.

## Data Availability

RNA-seq data generated in this manuscript were deposited in GEO (Gene Expression Omnibus) of NCBI under accession code GSE264394.

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
