# Peer review of "Examining Transcriptomic Alterations in Rat Models of Intracerebral Hemorrhage and Severe Intracerebral Hemorrhage"

_biomolecules, 2024, doi:10.3390/biom14060678_

Round 1

Reviewer 1 Report

Comments and Suggestions for Authors

In the rpesent study analysis of the transcriptome revealed the activation of specific pathways related to inflammation, immune response, and cell proliferation in rats- ICH model. These results indicate upregulated inflammatory and immune response, what is expected.Moreover, TNF signaling pathway, protein processing in the endoplasmic reticulum, MAPK signaling pathway, and Fc gamma R-mediated phagocytosis, is indicated as key in the pathogenesis of ICH.

Major comment: Please indicate the number of animals used, and what kind of control animals were used.

Reviewer 2 Report

Comments and Suggestions for Authors

Dear authors, while I have enjoyed reading your manuscript  and browsing through you complex figures, there are several potential issues that have raised questions regarding the present manuscript.

The methods section is rather superficial and does not offer the adequate information regarding the study lot as well as regarding inclusion criteria.

Line 77 the authors should mention how many animals were utilized as well as inclusion and exclusion criteria after the provocation of ICH.

Authors state in 176 that the hematoma volume is quantified in figure 1b but this must be further detailed in the text.

Lines 234 to 236 authors mention an astonishing number of genes however shouldn't they have focused throughout on a few dozen of them perhaps rather than just sheer numbers ? What scientific purpose would serve  such a generalization and quantifying strictly the number of genes ?

What was the benchmark for "severe" ICH, how did the authors quantify and differentiate ICH from SEVERE ICH ?

Comments on the Quality of English Language

English requires moderate corrections throughout. 

Round 2

Reviewer 2 Report

Comments and Suggestions for Authors

The authors have responded to my comments and the manuscript has improved. 

Comments on the Quality of English Language

English is fine overall.